# Clinical Characteristics and Prognosis of the Modified Probable *Pneumocystis jirovecii* Pneumonia in Korean Children, 2001–2021

**DOI:** 10.3390/children9101596

**Published:** 2022-10-21

**Authors:** Kyoung Sung Yun, Bin Anh, Sung Hwan Choi, Kyung Taek Hong, Jung Yoon Choi, Ki Wook Yun, Hyoung Jin Kang, Eun Hwa Choi

**Affiliations:** 1Department of Pediatrics, Seoul National University Children’s Hospital, Seoul 03080, Korea; 2Department of Pediatrics, Seoul National University College of Medicine, Seoul 03080, Korea; 3Seoul National University Cancer Research Institute, Seoul 03080, Korea

**Keywords:** *Pneumocystis jirovecii* pneumonia, children, risk factor

## Abstract

There are few data about *Pneumocystis jirovecii* pneumonia (PCP) in children, particularly in developed countries. This study investigated the clinical characteristics and prognosis of the clinical PCP in non-HIV-infected Korean children. Children with positive results for the staining and/or polymerase chain reaction (PCR) for *P. jirovecii* between 2001 and 2021 were identified. Patients were grouped into clinical PCP, which comprised proven and modified probable cases, and non-PCP groups. Modified probable PCP (mp-PCP) indicate the case which *P. jirovecii* was detected by conventional PCR rather than real-time PCR test. The differences in demographic and clinical characteristics were analyzed between the groups. A total of 110 pneumonia cases with positive results for *P. jirovecii* PCR and/or stain were identified from 107 children. Of these, 28.2% were classified as non-PCP, 12.7% of proven PCP, and 59.1% of mp-PCP. Compared with the non-PCP group, the mp-PCP group had a significantly higher rate of solid organ transplantation (3.2% vs. 24.6%), fever (58.1% vs. 76.9%), tachypnea (25.8% vs. 66.2%), dyspnea (48.4% vs. 83.1%), desaturation (48.4% vs. 80.0%), and bilateral ground-glass opacity on chest radiograph (19.4% vs. 73.8%). However, when the mp-PCP group was compared with the proven PCP group, there was no statistically significant difference. For children with clinical PCP, age under 5 years of age (odds ratio [OR] 10.7), hospital-onset (OR 6.9), and desaturation as initial symptom (OR 63.5) were significant risk factors for death in multivariable analysis. Modified probable PCP might reliably reflect true PCP in terms of patient’s demographic, clinical features, treatment response, and prognosis. Immunocompromised children with hospital-onset pneumonia who are younger than 5 years of age and have desaturation would be more cautiously and aggressively managed for survival through the screening for *P. jirovecii* by conventional PCR on appropriate lower respiratory specimens.

## 1. Introduction

*Pneumocystis jirovecii* pneumonia (PCP) causes substantial morbidity and mortality in immunocompromised patients, which include those infected with human immunodeficiency virus (HIV), patients receiving prolonged corticosteroid therapy, chemotherapy, and/or organ transplantation, patients with hematologic malignancy, primary immune deficiency, or severe malnutrition [1]. In the case of PCP, respiratory symptoms can rapidly develop and lead to intensive care unit (ICU) admission in children as well as the adult population [2,3]. However, few data are available on PCP in non-HIV infected children in developed countries at present. Furthermore, almost all studies only included proven PCP cases; thus, those studies had a very limited number of PCP cases, even for a long period of time [4,5].

The European Organization for Research and Treatment of Cancer and the Mycoses Study Group Education and Research Consortium (EORTC/MSGERC) recently updated the consensus definitions for *P. jirovecii* disease [6]. The diagnosis of proven PCP is based on the demonstration of *P. jirovecii* by microscopy using conventional or immunofluorescence (IF) staining in tissue or respiratory tract specimens, particularly bronchoalveolar lavage (BAL) fluid [7]. However, the sensitivity of staining is low and BAL is not a simple procedure, particularly in children. Therefore, PCP is often diagnosed with detection of *P. jirovecii* from induced sputum or transtracheal aspirate (IS/TTA) by using polymerase chain reaction (PCR). In this regard, EORTC/MSGERC defined probable PCP as the detection of *P. jirovecii* deoxyribonucleic acid (DNA) by quantitative real-time PCR (qPCR) in respiratory specimens and/or detection of β-d-glucan (BDG) in serum [7]. However, qPCR has not yet been widely distributed and validated for its cut-off value in South Korea. Therefore, in actual clinical practice in this country, it has been common to treat PCP with a modified diagnosis of probable PCP through conventional PCR results performed on IS/TTA. Thus, this study aimed to investigate the clinical features, prognosis, and risk factors of proven and modified probable PCP (mp-PCP) diagnosed by conventional PCR in Korean children at a single institution for the last 21 years.

## 2. Materials and Methods

From January 2001 to December 2021, patients aged 18 years old or younger who obtained positive results from the direct fluorescent antibody (DFA)/IF and/or conventional nested PCR test at Seoul National University Children’s Hospital were included. After reviewing medical records, only those with proven and mp-PCP were selected. For selected patients, information on demographics, clinical symptoms and signs, laboratory test results, prophylaxis and treatment of PCP, and prognosis were collected. During the study period, the DFA test was used until 2012, and the IF method was used thereafter for *P. jirovecii* stain. For PCR testing, the extracted DNA was subjected to a nucleic acid amplification process through nested PCR with oligonucleotide primer sets, targeting a large subunit of the mitochondrial ribosomal RNA gene, as previously described [8]. The presence of *P. jirovecii* was confirmed through electrophoresis using LabChip^®^ GX DNA Assay kit (Caliper Co., Newton, MA, USA). Respiratory samples included only BAL and IS/TTA. Beta-D glucan has been used intermittently in patients with suspected PCP since 2020, so information about the BDG test results was not collected in this study.

Proven and probable PCPs were defined according to EORTC/MSGERC criteria [7]. The definitions were based on the established triad of host factors, clinical features, and microbiological tests. The host factor was the use of therapeutic doses of ≥0.3 mg/kg prednisone equivalent for ≥2 weeks in the past 60 days and low CD4^+^ lymphocyte counts (observed or expected; <200 cells/mm^3^) induced by a medical condition, anticancer, anti-inflammatory, and immunosuppressive treatment. Clinical features consisted of two components: first, there symptoms and signs including fever and respiratory symptoms such as cough, dyspnea, or hypoxemia. The other was the radiologic pattern, which appears as bilateral and diffuse ground glass opacity (GGO) on X-ray, or mostly diffuse GGO on computed tomography (CT) scans, which typically has an upper lobe and perihilar predominance.

The diagnosis of proven PCP was based on clinical and radiologic criteria plus a demonstration of *P. jirovecii* by microscopy using conventional or IF staining in tissue or lower respiratory tract specimens. Probable PCP was defined by the presence of appropriate host factors and clinical-radiologic criteria, plus amplification of *P. jirovecii* DNA by qPCR in respiratory specimens and/or detection of BDG in serum, provided that another invasive fungal disease and a false-positive result can be ruled out [7]. In this study, we defined mp-PCP as a case in which *P. jirovecii* detection was performed by conventional nested PCR rather than qPCR while satisfying the criteria of existing probable PCP. Additionally, clinical PCP was defined as a term comprised of proven and mp-PCPs. Non-PCP was defined as a case with positive stain/PCR but not compatible with PCP in terms of host factors, clinical features, or radiologic findings.

The differences in demographic and clinical characteristics between proven PCP, mp-PCP, and non-PCP patients were analyzed with chi square test, Fisher’s exact test, and Kruskal–Wallis test. For the analysis of risk factor of mortality, multiple variable regression analysis was performed by including the variables with *p* < 0.1 in the univariate analysis as variables. All *p*-values were two-sided, and values below 0.05 were considered statistically significant. We performed all statistical analyses using SPSS software (v.25.0; IBM Corp., Armonk, NY, USA).

## 3. Results

### 3.1. Annual Numbers of P. jirovecii Detection and PCP Diagnosis

A total of 783 children underwent the *P. jirovecii* detection test (PCR (*n* = 1565), DFA (*n* = 410), and IF (*n* = 804)) during the study period, and 107 (13.7%) of them reported positive results in 110 cases. Among them, 31 (28.2%) cases were classified as non-PCP, and 14 (12.7%) cases of proven PCP and 65 (59.1%) cases of mp-PCP were identified. The main reasons for categorizing children with detected *P. jirovecii* into the non-PCP group were the presence of an alternative diagnosis and improved disease without PCP-specific treatment (*n* = 15), no compatible radiologic findings (*n* = 10) or host factors (*n* = 5), and another definitive diagnosis with non-compatible clinical presentation to PCP (*n* = 1). The number of occurrences by year of each case is shown in Figure 1. Until 2015–2017, the number of *P. jirovecii* detection test and diagnoses showed an upward trend. However, in 2018, there was a very large reduction in cases, despite a similar level of testing. Since then, the level has been gradually increasing again.

### 3.2. Clinical Characteristics of Children with mp-PCP

The median age of patients with mp-PCP was 7.4 years and the proportion of females was 47.7%. All but one had underlying disease. Solid organ transplant (SOT) recipient was the most common as 24.6%, followed by hematologic malignancy (HM) of 23.1% and hematopoietic cell transplant (HCT) recipient of 20.0%. As initial symptoms, fever and cough were present in 76.9% and 75.4%, respectively. Dyspnea was reported in 83.1% and desaturation was shown in 80.0%. Physical examination revealed rale in 29.2% and wheezing in 6.2%. Chest CT finding showed bilateral GGO pattern in 73.8% and pleural effusion (PE) in 7.7%. Hospital-onset PCP accounted for 36.9%. Twenty-nine percent (*n* = 19) of patients were receiving PCP prophylaxis within 30 days before the time of PCP diagnosis, and, among them, 42.1% (*n* = 8) received pentamidine rather than conventional trimethoprim/sulfamethoxazole (TMP/SMX) therapy. PCP-specific treatment was performed in 90.8% of patients with mp-PCP, TMP/SMX alone in 73.8%, and both TMP/SMX and pentamidine in 16.9%. The mortality rate was 27.7% (Table 1).

Compared with the non-PCP group, the mp-PCP group had a significantly higher rate of SOT (3.2% vs. 24.6%, *p* = 0.033) as underlying disease, fever (58.1% vs. 76.9%, *p* = 0.026), tachypnea (25.8% vs. 66.2%, *p* = 0.002), dyspnea (48.4% vs. 83.1%, *p* = 0.004), and desaturation (48.4% vs. 80.0%, *p* = 0.006) as initial symptom. There were significantly frequent findings of bilateral GGO pattern on chest CT in mp-PCP compared to non-PCP (73.8% vs. 19.4%, *p* < 0.001). When the mp-PCP group was compared with the proven PCP group, the bacterial culture positive rate was higher (23.1% vs. 7.1%), and the rate of bilateral GGO pattern (73.8% vs. 92.9%), usage rate of pentamidine for prophylaxis (12.3% vs. 21.4%) and PCP-specific treatment (16.9% vs. 42.9%) and the mortality rate (27.7% vs. 42.9%) were lower, but all these comparisons were statistically not significant (Table 1).

### 3.3. Co-Detected Respiratory Pathogens in Clinical PCP

Human rhinovirus (HRV) was the most common (15.2%) respiratory pathogen co-detected with *P. jirovecii* in clinical PCP, followed by cytomegalovirus (CMV) of 10.1%, respiratory syncytial virus (RSV) of 7.6%, and parainfluenza virus (PIV) of 7.6%. Among the bacteria, viridans *Streptococcus* (6.3%) was the most common, followed by *Acinetobacter baumannii* (3.8%) and *Staphylococcus aureus* (3.8%) (Figure 2). Although there were no significant differences, the codetection rate of respiratory pathogen was 7.1% in proven PCP, which tended to be lower than 23.1% in mp-PCP and 35.5% in non-PCP. Among children in mp-PCP group, CMV culture was positive in 12.3%, respiratory viruses were detected by multiplex RT-PCR in 35.4% and pyogenic bacteria was cultured in 23.1% (Table 1).

### 3.4. Factors Associated with Death in Clinical PCP

As a result of univariate analysis, the proportion of patients under 5 years of age, hospital-onset pneumonia, PCP prophylaxis, and combination of TMP/SMX and pentamidine was significantly higher in the death group than in the survival group (Table 2). In addition, multivariate analysis was performed by adding factors of PCP treatment, desaturation, which presents as an initial symptom, and chest CT findings. As a result, it was analyzed that age under 5 years old (odds ratio [OR] 10.7, 95% confidence interval [CI] 1.8–62.6), hospital-onset PCP (OR 6.9, 95% CI 1.5–32.3), and desaturation as initial symptom (OR 63.5, 95% CI 2.5–1645.5) were significant risk factors for death (Table 3). 

## 4. Discussion

This study investigated the epidemiology, clinical characteristics, and prognostic factors of PCP clinically diagnosed in children aged 18 years old or younger at a single institution for the last 21 years. Modified probable PCP was defined in this study as a positive result of *P. jirovecii* using conventional PCR, which has been widely used in actual clinical practice in South Korea. This disease entity was comparable to the proven PCP in terms of patient’ demographics, clinical features, laboratory test results, treatment response, and prognosis. Risk factors for mortality in these immunocompromised children with clinical PCP were identified to be younger age, hospital-onset PCP, and desaturation at presentation.

*Pneumocystis jirovecii* cannot be cultured, and there is no reliable single diagnostic test in PCP diagnosis. The tests recommended by EORTC/MSGERC, IF stain and qPCR, have limitations in practical application due to their low sensitivity, compliance, and universality [9]. Particularly for qPCR, it might be important to set an appropriate target gene and cutoff value to distinguish between colonization and active infection of *P. jirovecii* [10,11]. For more than 10 years, qPCR has had an important place in the diagnostic strategy for PCP. Recently, serum BDG has been proposed as method to improve diagnostic accuracy for PCP. The sensitivity and specificity in PCP diagnosis can be enhanced when the cut-off value of qPCR is adequately set and used in combination with other tests [12]. However, during the last 21 years in SNUCH, only DFA/IF and nested conventional PCR were used for *P. jirovecii* detection, and qPCR and BDG were introduced in 2022. The number of proven PCP cases was very small, but it is possible that most children with PCP were diagnosed by conventional PCR alone and successfully treated.

Although the incidence of PCP varies according to the host factors, children with high-risk ALL, high-grade non-Hodgkin lymphomas are at the highest risk. Children with Hodgkin’s lymphoma and children undergoing HCT are also at high risk for developing PCP. Overall, 0.63% of allogeneic and 0.28% of autologous recipients of their first HCT developed PCP [13]. In the current study, all but two patients with clinical PCP had underlying diseases related to immunocompromised status. HCT, SOT, and hematologic malignancy consisted of the majority of the underlying conditions in our patients. However, PID patients were relatively common in the non-PCP group. Therefore, we could more reliably consider PCP in children with hematologic malignancy and/or organ transplantation, whereas we might as well apply more stringent criteria for PCP diagnosis in children with PID, when an immunocompromised children showed a clinical presentation of PCP.

Additionally, in children, typical clinical features of PCP are known to include dry cough, insidious dyspnea, low-grade fever, tachypnea, hypoxia, and bilateral GGO on chest radiograph [3,13]. In the current study, most proven PCPs showed these clinical features, and the mp-PCP group also had very similar clinical features to the proven PCP group, while they were very different from those of the non-PCP group. This finding indicates that the mp-PCP diagnosis, which was temporarily defined in the current study, would reliably include many children with PCP, who could be missed unless PCP PCR is performed. For the clinical purpose, particularly in immunocompromised patients, the diagnostic method should have a higher sensitivity. Therefore, the mp-PCP definition can be used for children in situations where the reliable qPCR test is not available.

In this study, although non-PCP was classified based on the EORTC/MSGERC criteria, since these children underwent work-up for PCP based on the clinical speculation, they might be suspected to be immunosuppressed and/or have unusual clinical and radiologic findings of pneumonia. Therefore, demographic features of children and clinical characteristics of pneumonia in non-PCP group were largely comparable to those in the clinical PCP group. The positive results of *P. jirovecii* PCR/stain in the non-PCP group might be derived from the airway colonization of *P. jirovecii* or false positivity of the tests. It might have an advantage as a control group in terms of controlling confounding factors in comparison analyses between PCP groups. However, it is possible that some children with actual PCP could be misclassified into the non-PCP group in situations where we do not have gold-standard diagnostic methods with high sensitivity and specificity. Nevertheless, non-PCP group showed some differentiated clinical characteristics, such as incidence of respiratory distress and C-CT finding, compared to mp-PCP and proven PCP groups.

In the current study, CMV, as well as usual respiratory viruses and bacteria, were frequently co-detected with *P. jirovecii* from lower respiratory tract samples. CMV might be reactivated and cause lung infection from children with an immunocompromised status, in which PCP can occur [14]. Co-infection with CMV would be associated with more pronounced T-cell immunosuppression and a risk factor for severe disease and poor outcomes in patients with PCP [15,16]. However, in the current study, CMV was not a significant risk factor for mortality in children with clinical PCP, although the small sample size should not confirm this finding.

The mortality rate of children with clinical PCP in this study was 30.4%, which was similar to previous studies on proven PCP [4,17,18]. Previous studies reported that high respiratory and/or pulse rate, elevated C-reactive protein and/or lactate dehydrogenase level, need for mechanical ventilation, hypoxia, combined bacteremia, and preexisting chronic lung disease were more common in mortality cases among adult patients with PCP [19,20]. Although we could not evaluate all these factors for risk analysis in this study, we found that a younger age, hospital-onset pneumonia, and desaturation as initial symptom were the independent risk factors of mortality in children with clinical PCP. These factors would reflect the host’s lower immune and/or worse clinical status at the time of development of PCP, and sever initial presentation. We may use these risk factors to initiate an adequate targeted antimicrobial therapy in children with clinical diagnosis of PCP.

This study has several limitations. First, due to the lack of pediatric studies on a large number of children with PCP, all the definitions and the interpretation of premises and results were based on the studies and guidelines focused on adults. Particularly in respiratory infection, the pathophysiology of disease and available diagnostic methods might differ between children and adults, so more careful interpretation of the findings should be warranted. Second, because we have collected and analyzed data from children with *P. jirovecii* detected by conventional PCR rather than qPCR, which is officially recommended as a diagnostic tool of probable PCP, some patients could not have true PCP. However, we showed the clinical usefulness and adequacy of the mp-PCP in this study. This could be reinforced with biomarkers for *P. jirovecii* such as BDG in the near future. Third, we only collected mortality data as the outcome. The need for mechanical ventilation and ICU admission would be valuable for analyzing risk factors for poor outcomes and/or severe disease. However, a large proportion of patients were admitted to the ICU with mechanical ventilation at the time of development of PCP; therefore, we thought it would be difficult to adequately evaluate these factors for risk assessment from PCP in this retrospective design. We may need a large-scale prospective multicenter study. Lastly, this was a single-institution study with a relatively inhomogeneous population (all children were HIV-negative but had several underlying pathologies); thus, the results in this study could not be generalized to other settings.

## 5. Conclusions

The mp-PCP, which is defined with *P. jirovecii* detection in patients with classic host factors and clinical features of PCP, might be a good, alternative diagnostic criteria for PCP when qPCR is not available or a reliable cutoff level is not set. In immunocompromised young children with hospital-onset pneumonia showing desaturation as the main initial symptom, the disease could be managed promptly and aggressively based on these new criteria for clinical PCP in children.

## Figures and Tables

**Figure 1 children-09-01596-f001:**
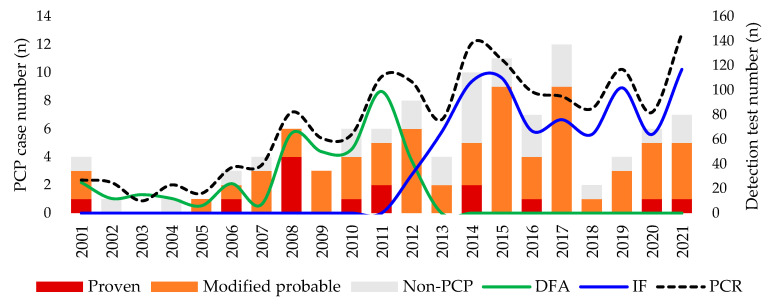
Annual number of *Pneumocystis jirovecii* testing performed and clinical PCP diagnosis. PCP, *Pneumocystis jirovecii* pneumonia; *n*, number; DFA, direct fluorescent antibody; IF, immunofluorescence; PCR, polymerase chain reaction.

**Figure 2 children-09-01596-f002:**
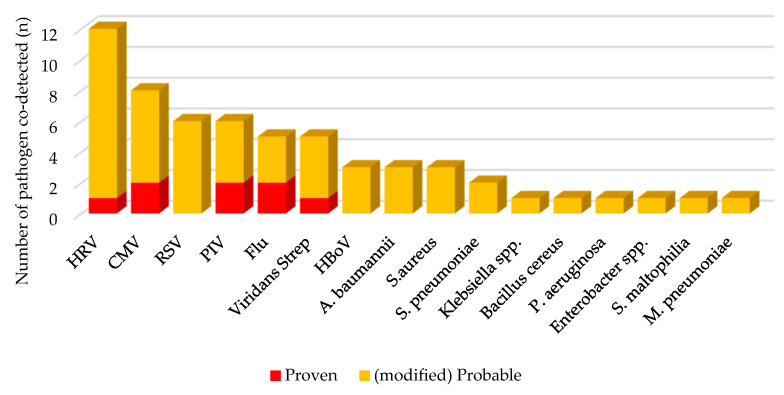
Frequency of respiratory pathogens co-detected in children with clinical PCP. PCP, *Pneumocystis jirovecii* pneumonia; *n*, number; HRV, human rhinovirus; CMV, cytomegalovirus; RSV, respiratory syncytial virus; PIV, parainfluenza virus; Flu, influenza virus; Strep, *Streptococcus*; HBoV, human bocavirus; *A. baumannii*, *Acinetobacter baumannii*; *S. pneumoniae*, *Streptococcus pneumoniae.*; spp., species; *P. aeruginosa, Pseudomonas aeruginosa; S. maltophilia, Stenotrophomonas maltophilia; M. pneumoniae, Mycoplasma pneumoniae*.

**Table 1 children-09-01596-t001:** Clinical characteristics of children with clinical PCP, *n* (%).

		Proven PCP (*n* = 14)	mp-PCP (*n* = 65)	Non-PCP (*n* = 31)	*p*-Value
Age (y)	Median (IQR)	5.5 (1.4–11.1)	7.4 (2.4–12.8)	4.7 (0.9–12.0)	0.173
	<5 YO	6 (42.9)	22 (33.8)	17 (54.8)	0.146
Sex	Female	6 (42.9)	31 (47.7)	17 (54.8)	0.712
Underlying medical condition	HSCT	3 (21.4)	13 (20.0)	4 (12.9)	0.662
	SOT	2 (14.3)	16 (24.6)	1 (3.2)	0.033 *
	HM	2 (14.3)	15 (23.1)	3 (9.7)	0.260
	SC	2 (14.3)	7 (10.8)	4 (12.9)	0.911
	PID	1 (7.1)	7 (10.8)	8 (25.8)	0.104
	None	1 (7.1)	1 (1.5)	3 (9.7)	0.178
Symptom/sign	Fever	13 (92.9)	50 (76.9)	18 (58.1)	0.026 *
	Cough	9 (64.3)	49 (75.4)	19 (61.3)	0.437
	Sputum	7 (50.0)	31 (47.7)	11 (35.5)	0.527
	Tachypnea	8 (57.1)	43 (66.2)	8 (25.8)	0.002 *
	Dyspnea	9 (64.3)	54 (83.1)	15 (48.4)	0.004 *
	Desaturation	11(78.6)	52 (80.0)	15 (48.4)	0.006 *
Breath sound	Rale	4 (28.6)	19 (29.2)	11 (35.5)	0.746
	Wheezing	0 (0.0)	4 (6.2)	9 (29.0)	0.318
Respiratory pathogen co-detected	CMV culture	2 (14.3)	8 (12.3)	2 (6.5)	0.695
	Virus RT-PCR	5 (35.7)	23 (35.4)	13 (41.9)	0.638
	Bacterial culture	1 (7.1)	15 (23.1)	11 (35.5)	0.103
C-CT findings	Bilateral GGO	13 (92.9)	48 (73.8)	6 (19.4)	<0.001 *
	Pleural effusion	2 (14.3)	5 (7.7)	3 (9.7)	0.675
Onset	Hospital onset	7 (50.0)	24 (36.9)	14 (45.2)	0.566
PCP Px	Yes	4 (28.6)	19 (29.2)	7 (22.6)	0.786
	PMD	3 (21.4)	8 (12.3)	3 (9.7)	0.475
PCP Tx	Yes	14 (100.0)	59 (90.8)	13 (41.9)	<0.001 *
	TMP/SMX only	8 (57.1)	48 (73.8)	11 (35.5)	0.119
	TMP/SMX + PMD	6 (42.9)	11 (16.9)	1 (3.2)	0.060
Outcome	Death	6 (42.9)	18 (27.7)	8 (25.8)	0.470

PCP, *P. jirovecii* pneumonia; *n*, number; mp-PCP, modified probable PCP; y, year; IQR, interquartile range; YO, years-old; HSCT, hematopoietic stem cell transplantation; SOT, solid organ transplantation; HM, hematologic malignancy; SC, solid cancer; PID, primary immune deficiency; CMV, cytomegalovirus; RT-PCR, reverse transcriptase polymerase chain reaction; C-CT, chest computerized tomography; GGO, ground glass opacity; Px, prophylaxis; Tx, treatment; PMD, pentamidine; TMP/SMX, trimethoprim/sulfamethoxazole. * *p* < 0.05 in comparison for three groups.

**Table 2 children-09-01596-t002:** Univariate analyses for risk factors of mortality in clinical PCP, *n* (%).

		Death (*n* = 24)	Survival (*n* = 55)	*p*-Value
Age (y)	<5 YO	13 (54.2)	15 (27.3)	0.022 *
Sex	Female	9 (37.5)	28 (50.9)	0.272
Underlying medical condition	HCT	9 (37.5)	7 (12.7)	0.012 *
	SOT	2 (8.3)	16 (29.1)	0.047 *
	HM	5 (20.8)	12 (21.8)	1.000
	SC	3 (12.5)	6 (10.9)	1.000
	PID	2 (8.3)	6 (10.9)	1.000
	None	2 (8.3)	0 (0.0)	0.900
Symptom/sign	Fever	20 (83.3)	43 (78.2)	0.532
	Cough	15 (62.5)	43 (78.2)	0.536
	Sputum	10 (41.7)	28 (50.9)	0.549
	Tachypnea	14 (58.3)	37 (67.3)	0.776
	Dyspnea	19 (79.2)	44 (80.0)	0.496
	Desaturation	22 (91.7)	41 (74.5)	0.055
Breath sound	Rale	9 (37.5)	14 (25.5)	0.197
	Wheezing	3 (12.5)	6 (10.9)	0.713
Respiratory pathogenco-detected	CMV culture	1 (4.2)	9 (16.4)	0.143
	Virus RT-PCR	8 (33.3)	20 (36.4)	0.781
	Bacterial culture	7 (29.2)	9 (16.4)	0.291
C-CT findings	Bilateral GGO	14 (58.3)	47 (85.5)	0.087
	Pleural effusion	4 (16.7)	3 (5.5)	0.080
Onset	Hospital onset	15 (62.5)	16 (29.1)	0.005 *
PCP prophylaxis	Yes	13 (54.2)	10 (18.2)	0.001 *
	PMD	5 (20.8)	6 (10.9)	0.414
PCP treatment	Yes	20 (83.3)	53 (96.4)	0.066
	TMP/SMX only	11 (45.8)	45 (81.8)	0.007 *
	TMP/SMX + PMD	9 (37.5)	8 (14.5)	0.007 *

PCP, *P. jirovecii* pneumonia; *n*, number; y, year; YO, years old; HSCT, hematopoietic stem cell transplantation; SOT, solid organ transplantation; HM, hematologic malignancy; SC, solid cancer; PID, primary immune deficiency; CMV, cytomegalovirus; RT-PCR, reverse transcriptase polymerase chain reaction; C-CT, chest computerized tomography; GGO, ground glass opacity; PMD, pentamidine; TMP/SMX, trimethoprim/sulfamethoxazole; * *p* < 0.05.

**Table 3 children-09-01596-t003:** Multivariable analysis for mortality in children with clinical PCP, *n* (%).

		Death (*n* = 24)	Survival (*n* = 55)	OR	95% CI
Age (y)	<5 YO	13 (54.2)	15 (27.3)	10.660 *	1.816–62.570
Underlying condition	HSCT	9 (37.5)	7 (12.7)	5.861	0.755–45.524
Onset	Hospital	15 (62.5)	16 (29.1)	6.941 *	1.491–32.308
Symptom/sign	Desaturation	22 (91.7)	41 (74.5)	63.547 *	2.454–1645.517
C-CT finding	Bilateral GGO	14 (58.3)	47 (85.5)	0.976	0.149–6.406
	Pleural effusion	4 (16.7)	3 (5.5)	8.026	0.754–85.413
PCP prophylaxis	Yes	13 (54.2)	10 (18.2)	5.294	0.770–36.386
PCP treatment	TMP/SMX + PMD	9 (37.5)	8 (14.5)	3.472	0.687–17.533

PCP, *P. jirovecii* pneumonia; *n*, number; OR, odds ratio; CI, confidential interval, year; YO, years-old; HSCT, hematopoietic stem cell transplantation; C-CT, chest computerized tomography; GGO, ground glass opacity; PMD, pentamidine; TMP/SMX, trimethoprim/sulfamethoxazole; * *p* < 0.05.

## Data Availability

The data that support the findings of this study are available from the corresponding author, upon reasonable request.

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
