# Peer review of "Clinical Characteristics and Prognosis of the Modified Probable Pneumocystis jirovecii Pneumonia in Korean Children, 2001–2021"

_children, 2022, doi:10.3390/children9101596_

Round 1

Reviewer 1 Report

In this article by Yun and colleagues, the authors report clinical characteristics and prognosis for clinical Pneumocystis jirovecii pneumonia (PCP) in non-HIV infected Korean children. In this study, authors used a modified definition for probable PCP by integrating results of conventional PCR instead of quantitative PCR (EORTC/MSGERC criteria, DOI: 10.1093/cid/ciz1008).

The main interest of this manuscript lies in the original current data provided on PCP in children, for which data in the literature are currently very limited.

Unfortunately, I have many reservations and suggestions regarding the structure/format of the manuscript, the design of the study, and the clarity of the discussion, as outlined below.

General, design and clarity:

- The main point of this study is not emphasized enough throughout the text. It is important to note at the outset how little data there is in the literature about PCP in children. The introduction needs to be more focused on the population of interest here, children, and although the data are limited, I think there is some specific references to PCP in children to add (lines 31-37).  In this way, the discussion can be more relevant and impactful with respect to the introduction and the more specific references provided.

- The authors point out the lack of validation of qPCR for the diagnosis of PCP (lines 50-52). I do not agree with this sentence, which is not supported by any reference. For more than 10 years now, qPCR has an important place in the diagnostic strategy for PCP. It is also in the official recommendations as quoted by the authors (EORTC/MSGERC 2021 criteria). The authors then decided to use a modified definition of probable PCP. Is there references to support this choice? If yes, they should be clearly stated. If not, the authors should clearly specify that this definition is their own. Then, to avoid any ambiguity throughout the text and the tables, it is important to specify each time whether we are talking about probable PCP (according to EORTC/MSGERC criteria) or modified probable PCP according to the authors' criteria.

- The meaning of "Clinical PCP" should also be specified in the body of the text, not just in the abstract, to clarify the discussion. Similarly, how should patients categorized as non-PCP be considered? Is it colonization? How do the authors explain the high mortality in this population?

- In the "Materials and Methods" section, the information given on the PCR used and the confirmation is not developed enough. If these methods are already explained in other papers, put the references.

- Part 3.2 is not clear. Many results are given without real link. It is necessary to restructure this part. Furthermore, the results on co-infections should be reserved for the part 3.3.

- On the data collected, do the authors have information on hospitalization in ICU and whether the patients were placed on mechanical ventilation. These elements would add to the discussion.

- The discussion deserves to be structured more neatly and clearly with the previously mentioned elements. It should be clearly identified if we are comparing modified probable PCP vs proven PCP, clinical PCP vs non-PCP...

In addition, there are many inaccuracies/misses in the manuscript:

- P. jirovecii and other microorganisms not italicized

- Inconsistent font (example: lines in table 1...)

- Extra spaces (line 16, line 184...)

- Awkward separation of "P." and "jirovecii" on two lines (example: line48-49)

- Missing punctuation (example: line 244)

- Explanation of abbreviations and acronyms missing in the text: HAP in the abstract, CAP line 210 and in the captions of tables/schemes/figures

- Lack of homogeneity in the chosen terms (example: Non-PCP or No PCP?)

- Lack of numerous references or references that are too old (line 45, lines 46-47, lines 52-54, line 72, line 172, line 187, line 209, line 215-219) or inadequate references (line 180)

- Wording to be revised: lines 190-193, lines 201-204, lines 230-231 ...

- Variability of conjugation tenses used in the "Materials and Methods" section

- No reference to Scheme 1 in the text

- Title of figure 1 does not match the content (Proven, modified probable and Non-PCP

- Possible confusion between rows in Table 1. What does the p-value refer to? 

Author Response

Thank you very much for the comments.

We revised our manuscript and we think it becomes better to be published.

Please find an attached file of our revised manuscript.

Reviewer 2 Report

The manuscript Clinical characteristics and prognosis of the modified probable

Pneumocystis jirovecii pneumonia in Korean children, 2001-2021is a retrospective study of children who received positive results from staining and/or polymerase chain reaction (PCR) for P. jirovecii at Seoul National University Children’s Hospital between 2001 and 2021. Patients were grouped into 3 clinical groups: a) children with proven, b) children with modified probable PCP, and c) children without PCP. All children were HIV negative. Modified probable PCP included children in which P. jirovecii was detected by conventional PCR rather than real-time quantitative PCR. The authors show that modified probable PCP likely reflects true PCP infection in terms of patients’ demographics, clinical features, treatment response, and prognosis. Immunocompromised children who were <5 years old, had desaturation as an initial finding, and who were diagnosed with hospital-onset clinical PCP had higher mortality requiring more cautious management.

            The manuscript is interesting but needs moderate revisions from a native English speaker. In its current form, it is relatively difficult to follow and understand. The authors use numerous abbreviations without explaining them at first appearance. For example, UMC (underlying medical condition), HRV (human rhinovirus) or PIV (parainfluenza virus) are not explained, and there are other similar abbreviations that are not analyzed. Please, change the word Bactrim with trimethoprim/ sulfamethoxazole (TMP/SMX).

            In addition, the authors should emphasize the weaknesses of their study, i.e., the fact that it is a single-institution study and with a relatively inhomogeneous population (all children were HIV-negative but had several underlying pathologies).

Author Response

(The authors gave the same response as above.)

Round 2

Reviewer 1 Report

  • The corrections made seem appropriate to me